# Current Challenges in the Diagnosis of Progressive Neurocognitive Disorders: A Critical Review of the Literature and Recommendations for Primary and Secondary Care

**DOI:** 10.3390/brainsci13101443

**Published:** 2023-10-10

**Authors:** Chiara Abbatantuono, Federica Alfeo, Livio Clemente, Giulio Lancioni, Maria Fara De Caro, Paolo Livrea, Paolo Taurisano

**Affiliations:** 1Department of Translational Biomedicine and Neuroscience (DiBrain), University of Bari “Aldo Moro”, 70121 Bari, Italy; chiara.abbatantuono@uniba.it (C.A.); livio.clemente@uniba.it (L.C.); giulio.lancioni@uniba.it (G.L.); maria.decaro@uniba.it (M.F.D.C.); 2Department of Education, Communication and Psychology (For.Psi.Com), University of Bari “Aldo Moro”, 70121 Bari, Italy; federica.alfeo@uniba.it; 3Lega F D’Oro Research Center, 60027 Osimo, Italy; 4Villa Anita, SP22, 70038 Terlizzi, Italy; paololivrea@email.it

**Keywords:** neurocognitive disorders, cognitive impairment, stadial progression, diagnostic criteria

## Abstract

Screening for early symptoms of cognitive impairment enables timely interventions for patients and their families. Despite the advances in dementia diagnosis, the current nosography of neurocognitive disorders (NCDs) seems to overlook some clinical manifestations and predictors that could contribute to understanding the conversion from an asymptomatic stage to a very mild one, eventually leading to obvious disease. The present review examines different diagnostic approaches in view of neurophysiological and neuropsychological evidence of NCD progression, which may be subdivided into: (1) preclinical stage; (2) transitional stage; (3) prodromal or mild stage; (4) major NCD. The absence of univocal criteria and the adoption of ambiguous or narrow labels might complicate the diagnostic process. In particular, it should be noted that: (1) only neuropathological hallmarks characterize preclinical NCD; (2) transitional NCD must be assessed through proactive neuropsychological protocols; (3) prodromal/mild NCDs are based on cognitive functional indicators; (4) major NCD requires well-established tools to evaluate its severity stage; (5) insight should be accounted for by both patient and informants. Therefore, the examination of evolving epidemiological and clinical features occurring at each NCD stage may orient primary and secondary care, allowing for more targeted prevention, diagnosis, and/or treatment of both cognitive and functional impairment.

## 1. Introduction

Major neurocognitive disorder, previously referred to as dementia, is a clinical condition entailing a significant decline in patients’ cognitive and daily living performances [1]. Screening for early symptoms of progressive, cognitive, and functional impairment would allow the use of early, more effective intervention to support patients and their caregivers [2]. Unfortunately, diagnostic criteria and tools currently available to detect subclinical, preclinical, and/or prodromal stages of neurocognitive disorders (NCDs) are not sufficiently developed and require further research work before one can rely on them as dependable means. The lack of harmonized neuropsychological assessment protocols [3] and the use of multiple measures for determining the onset/presence of impairment emphasize the need for more sensitive, specific, and culture-fair assessment tools and strategies [4]. In view of this challenging situation, healthcare professionals are charged with the daunting task of having to recognize the transitional stages between healthy and pathological aging to discriminate between individuals affected by dementing disorders and those who meet the diagnostic criteria for cognitive impairment no dementia (CIND) or mild cognitive impairment (MCI) [4]. 

Within any diagnostic approach, the professional will need to consider aging-related confounders and comorbidities, methodological deficiencies, and potential risks for clinical practice associated with “undetected” signs of dementia or cognitive impairment [5]. On the one hand, cognitive decline may advance over several years (or even decades) in the absence of overt clinical signs, making it difficult for the professional to recognize its development and the underlying pathophysiological processes [6]. On the other hand, assigning diagnostic labels to patients with subjective complaints or mild cognitive symptoms could prove to be a rushed and erroneous approach causing stress and anxiety for the patients and their caregivers [7] and leading to further/excessive testing and unnecessary interventions [8].

The present paper is an attempt to present and examine the stages that might define the time span leading from a condition of apparent wellbeing to an overt symptomatology of cognitive–functional impairment. Although the categorical diagnosis of NCDs represents the current diagnostic standard, it primarily relies on quantitative criteria addressing the severity of the neurocognitive impairment. Given NCD heterogeneity, this work focuses on delineating the progression of neurocognitive decline based on multiple models retrieved from the literature while also incorporating NCD manifestations that may arise at specific stages of decline. To this end, this review focuses on the stages of neurocognitive decline that can be identified as: (1) preclinical stage; (2) transitional stage; (3) prodromal or mild stage; (4) major NCD. As shown in Figure 1 and Figure 2, the paper also addresses CIND as a separate nosographic entity encompassing clinically mild or moderate conditions that, however, do not reach the threshold for a major NCD diagnosis. The proposed stages, which have been conceptualized based on evidence and recommendations reported in the literature, require further research to be adopted as an operational framework for clinicians.

## 2. Methods

To yield a stadial progression of neurocognitive decline, a review of the most widely used criteria for NCDs [1,9] and MCI [10,11,12] was conducted. The search was also implemented using the online databases PubMed and Scopus to investigate the neurocognitive stages characterized by signs and clinical manifestations preceding the onset of mild and/or major NCDs. In addition to considering their epidemiological and clinical features, the diagnostic criteria and stadial models for NCDs are also critically discussed in relation to their applicability in primary and secondary care. All materials are available in the Open Science Framework (OSF) at https://osf.io/84wnp/ (accessed on 10 September 2023).

## 3. Results

### 3.1. Preclinical Stage (i.e., Stage 1)

The preclinical stage is characterized by neuropathological hallmarks of a specific disease (e.g., Alzheimer’s disease, Parkinson’s disease, frontotemporal degeneration, vascular injury or disease) that do not translate into relevant or persistent cognitive symptoms. Research has focused on this phase to highlight potential risk factors of cognitive decline and the need to prevent the onset of NCD manifestations through primary prevention measures [13]. Among these measures, early- and late-life engagement in cognitive, physical, and social activities is deemed to be associated with greater cognitive maintenance, and this could explain why most individuals who test positive for β-amyloid may not develop any NCDs [14].

Preclinical Alzheimer’s disease (AD) consists of three distinct sub-stages that have been conceptualized based on in vivo studies [6]: (1) asymptomatic amyloidosis; (2) amyloidosis with “downstream” neurodegeneration; (3) amyloidosis with very mild cognitive and/or behavioral symptoms. Low cerebrospinal fluid β-amyloid protein 42 (CSF Aβ42) provides evidence of a neuropathological load that may be detected even before amyloidosis becomes visible on positron emission tomography (PET) imaging. Hence, isolated CSF Aβ positivity can be the first measurable sign of preclinical AD [15]. During that time, patients might start experiencing very mild neuropsychological alterations that require highly sensitive tools to be detected.

Overall, preclinical AD is prevalent in the elderly population. Considering a sample composed of 450 subjects aged more than 65 years and expected to be “cognitively normal”, 43% showed no alterations (sub-stage 0), 16% fell into asymptomatic amyloidosis (sub-stage 1), 12% showed signs of neurodegeneration (sub-stage 2), 3% reported cognitive/behavioral alterations (sub-stage 3), 23% featured negative Aβ but positive neurodegeneration markers, and the remaining 3% were unclassified [16]. After one year, about 43% of sub-stage 3 developed MCI or dementia [17]. This indicates that early cognitive symptoms should be subject to clinical monitoring within the first few months following their onset.

Biomarkers emerging in preclinical AD can also contribute to differential diagnosis, as Aβ deposition detected by Aβ PET imaging and/or by low CSF Aβ1-42 levels represents a prerequisite (i.e., a risk factor) for clinical subjects to be included within the “Alzheimer continuum”. However, the diagnosis of AD requires additional investigations through 18-F-fluorodeoxyglucose (FDG)-PET, structural MRI (sMRI), cortical Tau PET, and CSF biomarkers. Reduced glucose metabolism in specific brain regions (AD “signature”), temporal medial atrophy and regional cortical atrophy consistent with typical or atypical disease phenotypes, decreased Aβ1-42 combined with increased total Tau, and increased p-Tau in CSF all can support the diagnosis; the combination of different abnormalities depends on the spatiotemporal neuropathology trajectory over several years [18,19,20]. However, MRI findings show that neurodegeneration in atypical forms of AD may primarily cluster in brain areas other than medial temporal regions (e.g., posterior cortical or frontoparietal atrophy [21]).

Evidence of neurodegeneration markers (e.g., increased total CSF Tau; brain atrophy; glucose hypometabolism), in the absence of Aβ deposition demonstrated by Aβ cerebral PET and/or by low CSF Aβ1-42 levels, may instead characterize a wide range of clinical conditions called Suspected Non-Alzheimer Pathology (SNAP) [22]. It should be noted, however, that SNAP subjects may develop Aβ deposits over time [23], suggesting the possibility of becoming at risk or being affected by comorbid AD, the latter as indicated by the presence of increased Tau hyperphosphorylation. Different spatiotemporal trajectories of the p-Tau accumulation are known in AD and other tauopathies, the future availability of cerebral Tau PET being of great value for the differential diagnosis “in vivo” of these diseases [24].

### 3.2. Transitional Stage (i.e., Stage 2)

Neurodegenerative diseases, notably the most common forms of AD, may progressively impact on cognition, starting from early clinical features, signs, and prodromes associated with an increased risk for conversion to dementia [13]. During this stage, subjective experiences of cognitive impairment may occur and persist for at least six months, becoming a source of concern for patients and their caregivers. Subjective cognitive decline (SCD) may be perceived by the patient alone, without being noticed by further informants (e.g., caregivers, healthcare providers) through observable clinical signs [25]. Data from international cohort studies of aging indicate that almost a quarter of people aged over 60 years suffer from memory complaints, forgetfulness, and cognitive concerns [26] in the absence of informant-reported alterations in cognitive functioning and daily activities. Albeit self-experienced, SCD constitutes a risk factor for both MCI [26] and mild NCD [27], with a conversion rate of 6.67% cases per year [28]. Therefore, this condition arouses considerable clinical interest as it helps to clarify the gray zone between age-consistent changes and accelerated cognitive decline, allowing both patients and clinicians to intervene before NCD occurs [13].

Within the transitional stage, early yet persistent shifts in behavior may arise, which can be isolated or associated with slight cognitive issues (including SCD). Such symptoms do not meet the minimum criteria of severity and specificity to be included in a diagnostic cluster of psychiatric disorders; however, they may fall into the broad category of mild behavioral impairment (MBI) [29]. MBI affects 76.5% of people with SCD and 83.5% of people with mild cognitive impairment (MCI) [30]. From a longitudinal perspective, MBI can remain isolated, or can be followed by SCD or a mild NCD and subsequently evolve into a major NCD [31], thus representing a phenotype of transitional disease (stage 2). With specific reference to AD, isolated MBI may represent a phenotype of preclinical disease (stage 1) following asymptomatic amyloidosis [30].

### 3.3. Prodromal or Mild Stage (i.e., Stage 3)

The third stage (i.e., the prodromal or mild stage of NCD) is characterized by cognitive and functional alterations corroborated by objective neuropsychological indicators, such as cognitive test scores, which allow for the detection of mental state impairments and domain-specific deficits over the neurocognitive continuum. In contrast to major NCDs, such alterations do not interfere with daily routine. In order to classify an array of mild clinical conditions, the Diagnostic and Statistical Manual of Mental Disorders, Fifth Edition [1], has proposed the nosographic label of mild NCD in place of the MCI construct adopted for the previous two decades to discriminate between mild and severe patterns of cognitive decline [10]. The manual has set new diagnostic criteria that resemble those internationally proposed for MCI (i.e., acquired cognitive impairment with preserved independence in functional abilities [32]). The same core criteria have been maintained for the updated DSM-5-TR version [9]. Accordingly, clinicians are asked to assess cognitive functioning with specific reference to six cognitive domains: (1) complex attention; (2) executive function; (3) learning and memory; (4) language; (5) perceptual-motor function; and (6) social cognition. A selective, modest decline relating to one of these domains may be sufficient to allow diagnosis of mild NCD in self-reliant individuals [1].

In the case of minor cognitive symptoms, it is also advisable to consider the clinical presentation of MCI [33] as it encompasses diverse profiles of subthreshold alterations affecting dementia-free persons. Even if the “core” MCI criteria have remained unchanged, its clinical features have evolved and improved over fifteen years of research [11], reaching the current manifestations summarized in Table 1.

MCI sub-types have been operationalized based on the number and types of cognitive domains that may be found to be spared or impaired in MCI subjects. In particular, memory impairment can lead to two different MCI sub-types, consisting of amnesic single-domain MCI (a-MCI-sd) and amnesic multiple-domain MCI (a-MCI-md). If memory is spared, it is possible to discriminate between non-amnesic single-domain MCI (na-MCI-sd) and non-amnesic multiple-domain MCI (na-MCI-md) based on the same criterion. The highly variable MCI prevalence estimates mainly depend on the sample examined, according to the use of MCI macro-categories (i.e., a-MCI and na-MCI) or more specific diagnostic labels (e.g., a-MCI-sd). Indeed, the prevalence of MCI is still uncertain due to the heterogeneity of its diagnostic criteria and indices; however, there is agreement on its increased prevalence with advancing age, ranging from 2–5% (60 years of age) to 4–30% (over 90 years of age) [34]. The incidence of MCI is 6.37% cases per year in the age group between 70 and 89 years old and is higher in men (7.24%) than in women (5.73%) [35]. MCI progression has been extensively investigated through longitudinal studies, and its clinical patterns eventually fit a neuropsychological profile that largely overlaps with mild NCD, being prodromal to the onset of a major NCD. Indeed, about 5–10% of MCI subjects develop a major NCD [36], and conversion to dementia may also concern most people reverting from MCI to normal cognition [37].


The higher diagnostic specificity of mild NCD criteria (in terms of both number of impaired cognitive domains and etiologic hypotheses), compared to MCI criteria, results in different prevalence estimates regarding mild cognitive symptoms that may be smoothed through the harmonization of cognitive test scores for MCI [38]. According to both the aforementioned criteria, cognitive concerns can be reported by the patients themselves, or by reliable informants, and be supported by test results. With regard to this point, it is necessary to consider that patients’ awareness about their slight cognitive deficits may involve anxiety-driven worsening in neuropsychological performances [7]. In a number of cases, self-awareness might be absent (anosognosia), entailing profound repercussions on diagnostic processes as well as prognostic trends as these patients may pay the cost of delayed diagnosis. Unawareness has not been included among mild NCD and MCI criteria that consider self- and informant-reported complaints as equally important for diagnostic purposes, yet anosognosia is associated with progressive neuropathological processes, representing a risk factor for MCI conversion to dementia [11,39].

One should note here that specific therapies for prodromal AD have provided disappointing or uncertain results that may also depend on misdiagnosis. Considering recent advances in the diagnosis of AD, such as the new discovery about Aβ plaques being a consequence (and not a cause) of cognitive impairment [40], the correction of cognitive risk factors still emerges as a key strategy to prevent the onset of major NCD [36,41].
brainsci-13-01443-t001_Table 1Table 1Sub-types of mild cognitive impairment (MCI) and related prevalence.**Memory****Impaired** Amnestic MCI (a-MCI)**Spared** Non-amnestic MCI (na-MCI)**Other cognitive domains****Impaired**Amnestic multiple-domain MCI(a-MCI-md)**Spared**Amnestic single-domain MCI(a-MCI-sd)**Impaired**Non-amnestic Multiple-domain MCI (na-MCI-md)**Spared**Non-amnestic Single-domain MCI(na-MCI-sd)**Prevalence of each sub-type**[35] Roberts et al., 2012~70%~30%[42] Rapp et al., 201042.8%6.3%26.7%24.1%[43] Busse et al., 200626.2%22.4%9.3%41.9%Abbreviations: MCI = mild cognitive impairment; a-MCI = amnestic MCI; na-MCI = non-amnestic MCI; md = multiple-domain; sd = single-domain.


### 3.4. Major NCD (i.e., Stage 4)

The DSM-5 [1] has replaced the term dementia with the more comprehensive label of major NCD, redefining the number and type of impairments on which to base the diagnosis. Compared to the previous edition, the manual editions from 2013 onward no longer consider as mandatory the co-occurrence of memory decline and further acquired cognitive disturbances (e.g., executive and/or behavioral deficits) [1,9]. The most recent criteria established for major NCD are similar to those pertaining to mild NCD, except for disease severity that is greater in the first case, threatening daily and instrumental activities (as provided for in the DSM-IV). The presence of behavioral disturbances constitutes an additional specifier.

Given the lack of biomarkers for clinical differential diagnosis, the distinction between major and mild NCDs while patients are developing from early mild advanced disorder (late MCI) to initial major disorder is arbitrary. The differential process relies on the clinical and psychometric assessment of cognitive impairment and mostly depends on the tools and cut-off values adopted by clinicians. While test scores indicative of MCI generally fall within 1–1.5 standard deviation (SD), the cut-off proposed to discriminate between mild and major NCDs is conventionally set at 2 SDs below normative expectations [1,44]. The assessment of MCI severity can consist of annual cognitive screening that is, however, especially focused on memory impairment [45]. In view of the complexity of the neurocognitive decline, the Clinical Dementia Rating (CDR^®^) is one of the most used staging instruments to screen for patients’ degree of impairment across cognitive and functional domains [46] with the following criteria: 0 = normal; 0.5 = very mild dementia (only partially overlapping with diagnostic criteria for MCI and mild NCD); 1 = mild dementia; 2 = moderate dementia; 3 = severe dementia. Despite the wide use of the CDR in primary care, this tool provides a staging measure for senile dementia of Alzheimer’s type [47]. In addition to the CDR, the use of comprehensive neuropsychological test batteries is recommended to account for the features that characterize the clinical progression of NCDs [3], also considering that the clinical picture may remain stable or evolve, entailing different healthcare outcomes. As primary neurodegenerative diseases progressively lead to disability and increasing needs for clinical care, it is necessary to monitor dementing conditions through quanti-/qualitative approaches to determine the different manifestations and degrees of NCDs.

The survival rate following the diagnosis of major NCD due to a primary neurodegenerative disease is significantly lower than that in the non-clinical population [48]. Reduced life expectancies may depend on several risk factors (e.g., age of the onset; male gender; disease severity and sub-type; comorbidities; socio-demographic variables) [48,49]. The mean survival time for patients diagnosed with AD is higher, up to 7–12 years depending on the disease onset, compared to individuals with mild-to-severe dementia stages assessed through the CDR^®^ (approximately 3–3.5 years [50]). Patients suffering from vascular dementia (VaD) or Lewy body dementia (LBD) survive, in most cases, only up to 4 years after the diagnosis [48]. The reduced NCD mortality emerging from the last decade of research might be due to: (1) a greater control of risk factors, notably in high-income countries; (2) the overall effectiveness of secondary and tertiary prevention measures or therapies available for secondary neurodegenerative diseases. On the whole, reduced mortality and incidence may contribute to greater stability in NCD prevalence [51].

### 3.5. Cognitive Impairment No Dementia (CIND)

In addition to the aforementioned stages, it is also possible to contemplate further clinical entities (e.g., age-associated memory impairment (AAMI); age-associated cognitive decline (AACD); senescent forgetfulness; cognitive impairment no dementia (CIND)) [8,32] that still need to be operationalized through unambiguous indicators and markers. In particular, CIND has emerged as a widespread condition among subjects that do not reach the diagnostic threshold for major NCD [52]. Its prevalence estimates vary across studies and suggest that CIND may be unrelated to age, yet this broad clinical condition has been only partially considered in epidemiological research and is often confused with MCI and mild NCDs [34]. Given that CIND embraces the most disparate manifestations of cognitive impairment, except for major NCD, it is obvious that CIND represents an “umbrella” label expected to be more prevalent than MCI and major NCDs [53].

The phenomenology of CIND is so multifaceted that greater accuracy in defining the clinical characteristics and diagnostic criteria is required for this syndrome [52], considering patients’ conversion rates from preclinical and/or subclinical conditions into NCDs. Previous studies indicate that about 10% of subjects affected by CIND may evolve into major NCD, 8.2% of whom may develop AD-type NCD and 5.7% cerebrovascular disease [52]. The combined effect of CIND and physical frailty is predictive of NCD, and people presenting simultaneously these two conditions have a fivefold risk of developing dementia [54]. These data endorse a broad conceptualization of CIND encompassing patients at higher risk for developing major NCD [53]. Accordingly, CIND is also associated with further conditions that may underlie cognitive decline, such as internal diseases (e.g., diabetes; heart failure) (24%); stroke (15%); cerebral vasculopathy (10%); depression and other psychiatric conditions (5%); neurological diseases (e.g., PD; traumatic brain injuries) (5%); past and recent alcohol abuse (2%); and neurodevelopmental disorders (1%) [52]. The early diagnosis and detection of CIND and its comorbid factors, therefore, pose a challenge to health professionals committed to chronic disease self-management [55].

To advance the diagnosis and treatments for such diverse etiologies of cognitive impairment, systematization of the diagnostic criteria of the CIND has been proposed in the last decade [56] based on previous conceptualizations [57]. These novel criteria have eventually placed the syndrome in an intermediate stage of impairment, which is included between MCI and major NCD and involves objective impairment in cognition that is more severe than MCI but not indicative of loss of functional abilities [56]. As a result, CIND appears to be similar to multiple-domain MCI or mild NCD (based on standardized testing); still, its clinical features have scarcely been investigated and seem too vague and unspecific [32] to allow for a clear placement of CIND within the stadial progression of NCDs.

## 4. Discussion

Although not exhaustive, this review intends to provide scholars as well as clinicians with a dimensional framework that stems from neurophysiological and neuropsychological approaches to the progression of neurocognitive decline. To this end, we focused on evolving epidemiologic and clinical features of cognitive impairment to orient primary care and secondary care in the early and differential diagnosis of NCDs through an integrated perspective. The detection, prevention, and/or treatment of such syndromes are indeed accompanied by new challenges and discoveries that may fit the above-described stages (i.e., preclinical stage; transitional stage; prodromal or mild stage; major NCD) and related neurocognitive profiles.

While there is agreement about the prevalence of cognitive decline among the elderly population, the current diagnostic gold standard for NCD [1] adopts categorical criteria that may result in a restricted view on unspecified, preclinical, and/or transitional syndromes. Moreover, the literature concerning preclinical NCDs is still lacking highly sensitive biological and cognitive indices, beyond AD hallmarks, that could contribute to understanding the subtle conversion from asymptomatic disease to early neuropsychological symptoms [6,17], including MBI [30]. Further issues requiring careful investigation concern hyperphosphorylation mechanisms underlying tauopathies, given the profound impact on cognitive functioning that may derive from comorbid SNAP and AD [24]. Overall, focusing on the preclinical stage is necessary to advance primary prevention measures against dementia, also considering that patients’ Aβ positivity may result from cognitive decline [40]. Nevertheless, empirical findings are still anchored on the so-called “Alzheimer continuum” as amyloidosis frequently occurs among individuals aged 65 years and over [14]. The asymptomatic disease may entail subjective experiences of cognitive decline (SCD) that persist over time and lead, in turn, to MCI or mild NCD [13,26,27]. During the transition from covert to overt disease, cognitive and behavioral symptoms gradually become a source of concern for patients and eventually result in prodromal patterns of cognitive decline based on domain-specific evaluation. In particular, it is advisable to use protocols for the early assessment of multiple cognitive–behavioral indices that may guide clinicians in the differential diagnosis (e.g., synucleinopathies [58]; neuropsychiatric syndromes; neuropsychiatric onset of NCDs). It follows that, over the preclinical–prodromal continuum, primary and specialized care should account for overdiagnosis and overtreatment risks, in addition to the opportunities offered by recommended screening procedures. Interpreting an array of reversible or benign conditions as NCDs is indeed a threat to public health, and the ethical implications of such overestimation should be detailed in guidelines and health programs [8]. Further challenges for clinical practices arise from the need to discriminate anosognosic from insightful individuals to recognize dementia predictors before the emergence of functional alterations. If cognitive and behavioral symptoms progress to daily living impairment, professionals are advised to monitor the severity of the NCD regularly through multi-level assessment procedures [3]. It should also be noted that a considerable proportion of MCI or CIND individuals never develop clinically overt dementia. Early/mild conditions could remain stable or even show spontaneous remission, emerging as risk factors or frailty manifestations rather than pathological entities.

The models of NCDs reported in the literature hold a dynamic view of such conditions that still reveals ambiguous and unclear aspects underlying the current nosography for cognitive impairment. These aspects might delay and complicate the diagnostic process as it relies on diverse theoretical and methodological frameworks. Novel imaging techniques and CSF biomarkers have shown great utility in the detection of early NCD manifestations, yet not all the aforementioned protocols have been adopted and validated across outpatient settings, resulting in poor clinical translatability. Further limitations apply to the DSM-5, as the NCD cluster does not cover specific criteria for progressive conditions (e.g., a diagnostic framework for subthreshold profiles of impairment, the clinical significance of brain areas and functions affected by neurodegeneration over time, and testing criteria). Hence, the widespread adoption of stadial criteria may represent an opportunity to set or advance screening and treatment standards. This scope could be afforded through the achievement of evidence-based protocols and reliable indicators of NCD progression that can be spent in a broader range of clinical and research contexts, granting early and appropriate care throughout the neurocognitive continuum.

Overall, the present stadial framework for NCDs may complement existing categorical diagnoses to offer practical benefits in the context of both primary and specialized care. This approach promotes early intervention through the identification of established hallmarks and symptoms that may convert to dementia, remain stable, or revert to a status of cognitive–functional wellbeing over time. Although no consensus on CIND makes it even harder to predict the prognosis of different “no dementia” syndromes, some proactive strategies can be adopted to counter mild-to-major conversion. Based on epidemiological data on the potential remissions of mild symptoms up to the prodromal stage, it becomes crucial to target the modifiable factors of dementia (e.g., cardiovascular health and lifestyle choices) aimed at fostering patients’ functioning and quality of life. While medications and cognitive-enhancing drugs can be effective in managing specific symptoms [59,60], favorable outcomes in the early stages of the neurocognitive continuum may be partially ascribable to brain and cognitive reserves, which act as protective factors against both age- and disease-related decline [14,61,62,63]. To this end, the involvement of caregivers from the onset of very mild symptoms is key. Caregivers can indeed engage the patients in stimulating experiences as well as monitor the persistence and impact of symptoms that may become evident at neuropsychological follow ups. Their role in informal care, which is being increasingly recognized by institutional care teams, is equally important when patients’ functioning and insight begin to decline [64].

By categorizing cognitive decline into stages, the healthcare system can also allocate resources efficiently, directing support to individuals at higher risk from the preclinical phase and evaluating cases where undergoing invasive exams can be avoided and integrated treatment strategies [60,65,66,67,68] can be advised. Moreover, this framework contributes to overtreatment and stigma reduction by acknowledging cognitive impairment as a continuum, encouraging individuals to seek medical help earlier and participate actively in their care decisions, and mitigating concerns that usually occur when undergoing screening [68].

## 5. Conclusions

The recognition and monitoring of individuals at the stages from preclinical to overt dementia are essential for optimizing clinical efforts against neurocognitive decline. These strategies may allow healthcare professionals to assess the progression of cognitive decline stepwise, orient care plans to evolving patient needs, and provide timely interventions to enhance their quality of life. In view of the validation of further assessment techniques and dementia biomarkers, the adoption of a perspective that accounts for clinical–temporal progression can enrich the diagnostic systems and supportive measures currently in use, ultimately advocating wellbeing and independence for individuals experiencing neurocognitive symptoms.

## Figures and Tables

**Figure 1 brainsci-13-01443-f001:**
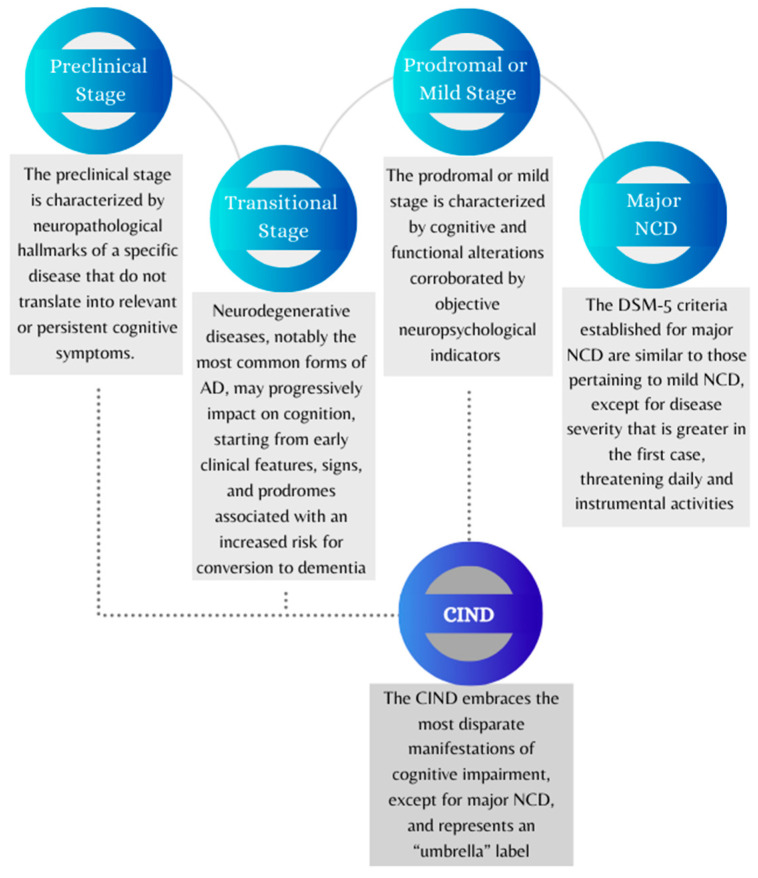
Graphical representation of four conventional stages of neurocognitive decline and cognitive impairment no dementia (CIND). The diagram shows the possible gradual progression of neurocognitive disorders from the preclinical stage to the onset of dementia. Abbreviations: AD = Alzheimer’s disease; CIND = cognitive impairment no dementia; DSM-5 = Diagnostic and Statistical Manual of Mental Disorders, 5th edition; NCD = neurocognitive disorder.

**Figure 2 brainsci-13-01443-f002:**
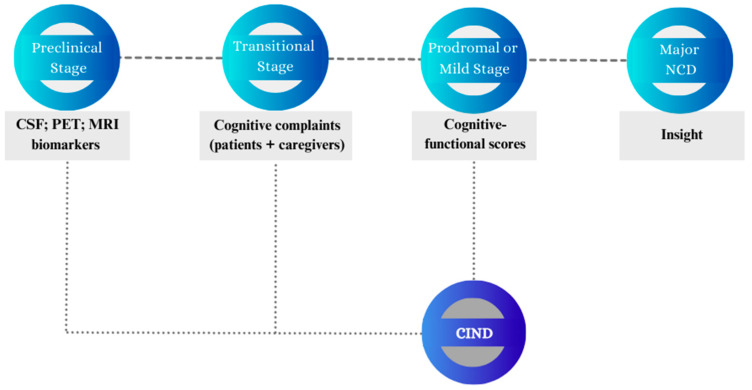
Graphical representation of the markers or indices covered by assessment procedures over the neurocognitive continuum. The figure summarizes selected measures and information that can be collected to support diagnosis and prognosis according to the most widely adopted classification models. Abbreviations: AD = Alzheimer’s disease; CSF = cerebrospinal fluid; CIND = cognitive impairment no dementia; MRI = Magnetic Resonance Imaging; NCD = neurocognitive disorder; PET = positron emission tomography.

## Data Availability

All materials are available on the Open Science Framework (OSF): https://osf.io/84wnp/ (accessed on 10 September 2023).

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
