# Peer review of "Current Challenges in the Diagnosis of Progressive Neurocognitive Disorders: A Critical Review of the Literature and Recommendations for Primary and Secondary Care"

_brainsci, 2023, doi:10.3390/brainsci13101443_

Round 1
Reviewer 1 Report
- The introduction is well-written and informative. It provides a good overview of the topic of the article, which is the stages of neurocognitive decline.
- There are a few minor issues with the introduction. For example, line 34 states that "the lack of harmonized neuropsychological assessment protocols" emphasizes the need for more sensitive, specific, and culture-fair assessment tools. However, it is not clear what is meant by "harmonized" in this context.
- Overall, the introduction is a good start to the article. It provides a clear and concise overview of the topic, and it sets the stage for the rest of the paper.
- In line 34, clarify what is meant by "harmonized" neuropsychological assessment protocols.
- In line 51, consider adding a sentence to explain why the paper focuses on the four stages of neurocognitive decline that are identified.
- Methods (lines 73-81): The methods section is well-organized and provides a clear overview of the search strategy and data analysis. The authors have also done a good job of justifying their choices and providing citations to support their claims.
- Results (lines 83-305): The results section is comprehensive and provides a detailed overview of the stages of neurocognitive decline. The authors have done a good job of summarizing the current literature and highlighting the key findings.
- Discussion (lines 307-365): The discussion section is well-written and provides a thoughtful analysis of the findings. The authors discuss the implications of their findings for clinical practice and future research.
-
Here are some specific suggestions for improvement:
- In line 155, the authors state that "MBI can be followed by an SCD or a mild NCD, and subsequently evolve into a major NCD". It would be helpful to provide more information about the rates of conversion from MBI to other stages of neurocognitive decline.
- In line 174, the authors state that "A selective, modest decline relating to one of these domains may be sufficient to allow diagnosis of mild NCD in self-reliant individuals". It would be helpful to provide more specific information about the severity of cognitive decline that is required for a diagnosis of mild NCD.
- Line 182-186: The authors could provide more detail on the clinical features of CIND (cognitive impairment no dementia). For example, they could discuss the specific cognitive domains that are typically affected in CIND (182), as well as the severity of the impairment (183).
- Line 196-199: The authors could also discuss the risk factors for progression from CIND to major NCD (196). This would help to inform clinicians about the patients who are most likely to benefit from early intervention (197).
- Line 208-211: Finally, the authors could discuss the potential benefits of using biomarkers to diagnose and monitor CIND (208). Biomarkers could help to identify patients who are at risk of progression (209), and could also be used to track the effectiveness of treatment (210).
- The discussion section of the paper is well-written and informative. The authors provide a balanced and nuanced assessment of the challenges and opportunities associated with the dimensional framework for the progression of NCD. They also make several important recommendations for future research and practice.
- Line 326: Discuss the potential benefits of using a stadial framework for NCD in terms of patient care and public health (e.g., identifying individuals at high risk of developing dementia and who may benefit from early intervention).
- Line 338: Discuss the challenges of implementing a stadial framework for NCD in clinical practice (e.g., need to develop standardized stadial criteria and train clinicians in their use).
- Line 357: Discuss the ethical implications of using a stadial framework for NCD (e.g., potential for overdiagnosis and overtreatment, need to protect the privacy and autonomy of individuals with NCD).
Author Response
LETTER 1.
We thank the Reviewers for the thorough comments that allowed us to improve our manuscript. Answers and changes within this letter have been reported in bold, while those in the paper have been marked in yellow.
R1: There are a few minor issues with the introduction. For example, line 34 states that "the lack of harmonized neuropsychological assessment protocols" emphasizes the need for more sensitive, specific, and culture-fair assessment tools. However, it is not clear what is meant by "harmonized" in this context. Overall, the introduction is a good start to the article. It provides a clear and concise overview of the topic, and it sets the stage for the rest of the paper. In line 34, clarify what is meant by "harmonized" neuropsychological assessment protocols.
We thank Reviewer 1 for this feedback. The “harmonized” neuropsychological protocols we mentioned are an explicit reference to Costa et al.’s guidelines (2017), cited in lines 48-51 and reported in the endnote bibliography. The document is entitled The need for harmonisation and innovation of neuropsychological assessment in neurodegenerative dementias in Europe (…), and critically discusses which and how many tests may be used for patients’ comprehensive assessment. The Authors highlight the need for objective evidence of cognitive impairment to make a diagnosis of neurocognitive disorder (NCD, as per DSM criteria), yet this need may encounter uncertainties in electing the most appropriate or suitable procedures within the clinical setting. As the article cited represents a consensus document, we adopted the original terminology proposed by Costa instead of “general acceptance” or “consensus” to refer to the multitude of assessment tools and techniques available to date.
R1: In line 51, consider adding a sentence to explain why the paper focuses on the four stages of neurocognitive decline that are identified.
We agree about the need to provide this explanation. Accordingly, we added two sentences (lines 58-73):
Although the categorical diagnosis of NCDs represents the current diagnostic standard, it primarily relies on quantitative criteria addressing the severity of the neurocognitive impairment. Given NCD heterogeneity, this work focuses on delineating the progression of neurocognitive decline based on multiple models retrieved from the literature while also incorporating NCD manifestations that may arise at specific stages of decline.
R1: Methods (lines 73-81): The methods section is well-organized and provides a clear overview of the search strategy and data analysis. The authors have also done a good job of justifying their choices and providing citations to support their claims. Results (lines 83-305): The results section is comprehensive and provides a detailed overview of the stages of neurocognitive decline. The authors have done a good job of summarizing the current literature and highlighting the key findings. Discussion (lines 307-365): The discussion section is well-written and provides a thoughtful analysis of the findings. The authors discuss the implications of their findings for clinical practice and future research.
We thank Reviewer 1 for shedding light on the main strengths of our manuscript.
R1: Here are some specific suggestions for improvement: in line 155, the authors state that "MBI can be followed by an SCD or a mild NCD, and subsequently evolve into a major NCD". It would be helpful to provide more information about the rates of conversion from MBI to other stages of neurocognitive decline.
We agree on the importance of adding data on MBI-to-dementia conversion. The DSM-5-TR emphasizes the importance of accounting for behavioral disturbances as specifiers for both mild and major NCD diagnoses. Even if it would be very interesting to explore all these aspects, we gathered much literature regarding the so-called “behavioral and psychological symptoms of dementia” (BPSD) and neuropsychiatric symptoms (NPS) in specific aetiologies of dementia, but evidence on MBI conversion is still quite scarce. Overall, NPS, even of mild entities, may be considered dementia hallmarks and be present at varying levels and rates in SCD and MCI populations (Dillon et al., 2013). However, most recent data on behavioral specifiers need further validation or may apply more to the clinical - and not the general -population. In view of the scarcity of literature on updated/established rates for MBI conversion, this topic is worth attention and further research.
References:
Dillon, C., Serrano, C. M., Castro, D., Leguizamón, P. P., Heisecke, S. L., & Taragano, F. E. (2013). Behavioral symptoms related to cognitive impairment. Neuropsychiatric disease and treatment, 1443-1455.
Ismail, Z., Smith, E. E., Geda, Y., Sultzer, D., Brodaty, H., Smith, G., ... & Area, I. N. S. P. I. (2016). Neuropsychiatric symptoms as early manifestations of emergent dementia: provisional diagnostic criteria for mild behavioral impairment. Alzheimer's & Dementia, 12(2), 195-202.
R1: In line 174, the authors state that "A selective, modest decline relating to one of these domains may be sufficient to allow diagnosis of mild NCD in self-reliant individuals". It would be helpful to provide more specific information about the severity of cognitive decline that is required for a diagnosis of mild NCD.
As for its previous editions, the DSM-5-TR provides professionals with a categorical system to make the diagnosis without setting severity thresholds for mild NCD. According to DSM criteria, mild NCD is characterized by objective evidence of impairment in at least one out of six cognitive domains. Such impairment is “modest” as opposed to that of major NCD/dementia which is “substantial”. Although these cognitive challenges have not been explicitly quantified in the diagnostic manual, patients with mild NCD typically maintain a level of functional independence (vs. major NCD).
The DSM substages only major NCD based on “current severity”, whereas the diagnosis of mild NCD is based on the presence of a cognitive impairment not interfering with daily life, including only aetiological and behavioral specifiers. It follows that even a deficit in one – no matter “which” – cognitive domain may be sufficient to make the mild NCD diagnosis as long as no exclusion criterion is met.
R1: Line 182-186: The authors could provide more detail on the clinical features of CIND (cognitive impairment no dementia). For example, they could discuss the specific cognitive domains that are typically affected in CIND (182), as well as the severity of the impairment (183).
We thank Reviewer 1 for this suggestion. A study in which CIND was operationalized based on cut-offs to neuropsychological tests revealed a higher likelihood of multidomain impairment in amnestic subtype CINDs, but it should be specified that, besides being not particularly recent, this study subdivided CIND participants a priori according to the type of domain impaired, so in this case neuropsychological patterns cannot be discriminative of CIND subtypes (Jacova et al., 2008). Other studies on the topic were also conducted during 2007-2008, adopting a data-driven model on specific samples (mainly stroke and mixed dementia), while a recent doctoral dissertation reported that “the sample demonstrated high rates of reversion and fluctuation in CIND status across years” and that “continued research on CIND stability is recommended to improve classification methodology”, supporting what we emphasized regarding the characterization of CIND as a sort of multidomain MCI that, however, also includes milder stages within it.
References:
Jacova, C., Peters, K. R., Beattie, B. L., Wong, E., Riddehough, A., Foti, D., ... & Feldman, H. H. (2008). Cognitive impairment no dementia–neuropsychological and neuroimaging characterization of an amnestic subgroup. Dementia and geriatric cognitive disorders, 25(3), 238-247.
McDowell, C. (2022). Longitudinal Patterns and Predictors of Cognitive Impairment Classification Stability (Doctoral dissertation).http://hdl.handle.net/1828/14115
R1: Line 196-199: The authors could also discuss the risk factors for progression from CIND to major NCD (196). This would help to inform clinicians about the patients who are most likely to benefit from early intervention (197). Line 208-211: Finally, the authors could discuss the potential benefits of using biomarkers to diagnose and monitor CIND (208). Biomarkers could help to identify patients who are at risk of progression (209), and could also be used to track the effectiveness of treatment (210).
We agree about these points and have, accordingly, included considerations on CIND-to-dementia conversion in the discussion part of the review (lines 399-408). Details have been fully reported in the final part of the present letter. However, “specific” conversion risk factors and markers have yet to be identified, as CIND can encompass transitional and prodromal features based on the approach adopted for its classification. Overall, we agree with other Scholars (e.g., Bradfield & Ames, 2019) about the fact that CIND can be useful to ensure early detection of dementia, but the absence of valid “operational criteria” may complicate its recognition, requiring a cautious interpretation of CIND conversion estimates.
According to the literature we reviewed, no standard has yet been set internationally. Hence, CIND diagnostic and prognostic biomarkers may coincide with those of MCI and mild NCDs or even “very mild” conditions, the signs of which depend on the known disease underlying the cognitive impairment (e.g., A/T/N in the case of AD-type impairments; dopamine metabolites for PD and parkinsonisms, etc.). However, even in the absence of validated prognostic models for CIND, considering its features can help the clinical understanding of a various array of non-dementing conditions over the neurocognitive continuum.
Reference:
Bradfield, N. I., & Ames, D. (2020). Mild cognitive impairment: narrative review of taxonomies and systematic review of their prediction of incident Alzheimer's disease dementia. BJPsych bulletin, 44(2), 67-74.
R1: The discussion section of the paper is well-written and informative. The authors provide a balanced and nuanced assessment of the challenges and opportunities associated with the dimensional framework for the progression of NCD. They also make several important recommendations for future research and practice.
Line 326: Discuss the potential benefits of using a stadial framework for NCD in terms of patient care and public health (e.g., identifying individuals at high risk of developing dementia and who may benefit from early intervention).
Line 338: Discuss the challenges of implementing a stadial framework for NCD in clinical practice (e.g., need to develop standardized stadial criteria and train clinicians in their use).
Line 357: Discuss the ethical implications of using a stadial framework for NCD (e.g., potential for overdiagnosis and overtreatment, need to protect the privacy and autonomy of individuals with NCD).
We thank Reviewer 1 for these suggestions regarding different key aspects to be included in the discussion part. We made an effort to integrate the Reviewer’s comments about the implication pertaining to mild-to-major impairments while addressing all the crucial challenges and opportunities stemming from NCD assessment for healthcare providers. All these additions have been reported as a comprehensive paragraph (lines 399-425):
Overall, the present stadial framework for NCDs may complement existing categorical diagnoses to offer practical benefits in the context of both primary and specialized care. This approach promotes early intervention through the identification of established hallmarks and symptoms that may convert to dementia, remain stable, or revert to a status of cognitive-functional wellbeing over time. Although no consensus on CIND makes it even harder to predict the prognosis of different “no dementia” syndromes, some proactive strategies can be adopted to counter mild-to-major conversion. Based on epidemiological data on the potential remissions of mild symptoms up to the prodromal stage, it becomes crucial to target the modifiable factors of dementia (e.g., cardiovascular health and lifestyle choices) aimed at fostering patients’ functioning and quality of life. While medications and cognitive-enhancing drugs can be effective in managing specific symptoms (Frederiksen et al., 2020; Pizzi et al., 2020), favorable outcomes in the early stages of the neurocognitive continuum may be partially ascribable to brain and cognitive reserve, which act as protective factors against both age- and disease-related decline (Livingston et al., 2020; Song et al., 2022; Stern et al., 2020, 2021). To this end, the involvement of caregivers since the onset of very mild symptoms is key. Caregivers can indeed engage the patients in stimulating experiences as well as monitor the persistence and impact of symptoms that may become evident at neuropsychological follow-ups. Their role in informal care, which is being increasingly recognized by institutional care teams, is equally important when patients’ functioning and insight begin to decline (Reckrey et al., 2021).
By categorizing cognitive decline into stages, the healthcare system can also allocate resources efficiently, directing support to individuals at higher risk since the preclinical phase, and evaluating cases where undergoing invasive exams can be avoided and integrated treatment strategies (Chowdhari et al., 2020; De Caro et al., 2022, Huang et al., 2021; Pini et al., 2020; Pizzi et al., 2020) can be advised. Moreover, this framework contributes to overtreatment and stigma reduction by acknowledging cognitive impairment as a continuum, encouraging individuals to seek medical help earlier and participate actively in their care decisions, and mitigating concerns that usually occur when undergoing screening (Gruters et al., 2021).
References:
Chowdhary, N., Barbui, C., Anstey, K. J., Kivipelto, M., Barbera, M., Peters, R., ... & Dua, T. (2022). Reducing the risk of cognitive decline and dementia: WHO recommendations. Frontiers in neurology, 12, 765584.
De Caro, M., Taurisano, P., Calia, C., & Abbatantuono, C. (2022). Profili e Modelli Neuropsicologici delle patologie neurodegenerative. Franco Angeli.
Frederiksen, K. S., Cooper, C., Frisoni, G. B., Frölich, L., Georges, J., Kramberger, M. G., ... & Waldemar, G. (2020). A European Academy of Neurology guideline on medical management issues in dementia. European journal of neurology, 27(10), 1805-1820.
Gruters, A. A., Christie, H. L., Ramakers, I. H., Verhey, F. R., Kessels, R. P., & de Vugt, M. E. (2021). Neuropsychological assessment and diagnostic disclosure at a memory clinic: A qualitative study of the experiences of patients and their family members. The Clinical Neuropsychologist, 35(8), 1398-1414.
Huang, X., Zhao, X., Li, B., Cai, Y., Zhang, S., Yu, F., & Wan, Q. (2021). Biomarkers for evaluating the effects of exercise interventions in patients with MCI or dementia: A systematic review and meta-analysis. Experimental Gerontology, 151, 111424.
Livingston, G., Huntley, J., Sommerlad, A., Ames, D., Ballard, C., Banerjee, S., ... & Mukadam, N. (2020). Dementia prevention, intervention, and care: 2020 report of the Lancet Commission. The Lancet, 396(10248), 413-446.
Pini, L., Manenti, R., Cotelli, M., Pizzini, F. B., Frisoni, G. B., & Pievani, M. (2019). Non-invasive brain stimulation in dementia: a complex network story. Neurodegenerative Diseases, 18(5-6), 281-301.
Pizzi, S. D., Granzotto, A., Bomba, M., Frazzini, V., Onofrj, M., & Sensi, S. L. (2020). Acting before; a combined strategy to counteract the onset and progression of dementia. Current Alzheimer Research, 17(9), 790-804.
Reckrey, J. M., Boerner, K., Franzosa, E., Bollens-Lund, E., & Ornstein, K. A. (2021). Paid caregivers in the community-based dementia care team: Do family caregivers benefit?. Clinical Therapeutics, 43(6), 930-941.
Stern, Y., Barnes, C. A., Grady, C., Jones, R. N., & Raz, N. (2019). Brain reserve, cognitive reserve, compensation, and maintenance: operationalization, validity, and mechanisms of cognitive resilience. Neurobiology of aging, 83, 124-129.
Stern, Y. (2021). How can cognitive reserve promote cognitive and neurobehavioral health?. Archives of Clinical Neuropsychology, 36(7), 1291-1295.
Song, S., Stern, Y., & Gu, Y. (2022). Modifiable lifestyle factors and cognitive reserve: A systematic review of current evidence. Ageing Research Reviews, 74, 101551.

Reviewer 2 Report
The Authors conducted a literature review on the neuropsychological characterisation of the continuum between asymptomatic neuropathological changes and full-blown dementia. It is a very interesting work that, with some modifications, will provide insight and useful suggestions to clinicians and researchers interested in the topic. I have some suggestions for the authors:
- Figure 1 would be more useful if it provided, for each category, elements that would highlight the differences between each in a clear and direct way for the reader (also in the form of a bulleted list);
- On the Methods section, I would advise the Authors to provide some more information on how the research was conducted: which keywords were searched? How many Authors took part in the data collection and discussion?
- Lines 96-97: Please specify what you mean by "very mild cognitive and/or behavioural symptoms". If the next stage (i.e. Transitional stage) includes patients with subjective problems that cannot be identified with commonly used neuropsychological tests, how are cognitive and/or motor symptoms identified and classified in this stage 1?
- Sentence line 97 to 99: Please add a citation.
- Sentence line 115 to 117: The Authors cite PET-tau as an additional criterion for the diagnosis of AD. According to which diagnostic criteria? Its use is often almost exclusively reserved for research protocols.
- Lines 118-119: Please add some details on what is meant by AD atypical variants and what their hallmarks are from the earliest stages, citing relevant literature on the subject.
- I advise the Authors, especially in the sections on the early stages, when the level of cognitive functioning often does not justify the use of invasive diagnostic means, to provide information on plasma biomarkers of the various neurodegenerative diseases (e.g. neurofilament light chains, plasma amyloid) and neurophysiological markers (e.g. diagnostic and/or prognostic approaches with NIBS techniques). For example, in the transition stage, which biomarkers are commonly found on plasma and CSF in different neuropathological conditions? How can they help predict evolution (or not) to later stages?
- Prodromal or mild stage (i.e., stage 3): at this stage, what are the neuropsychological information (including neuropsychological tests of specific utility) and biomarkers (CSF, plasma and PET) that can help predict evolution into the various forms of dementia?
- Line 224: please add the full name of the cited source as [1].
- Lines 259-261: the meaning of this sentence is not clear to me, please rephrase.
- Lines 264-265: mortality and incidence are two different epidemiological measures. Mortality may decrease due to better medical management of the patient, but the incidence (according to many epidemiological estimates and evaluations) of neurodegenerative dementias (especially AD) is likely to increase significantly over the next decades. I suggest that the Authors justify this sentence by citing the relevant bibliography or rephrase this sentence.
- In the Discussion, I suggest adding specific advice on the optimal course of action in each stage examined (e.g. preventive measures, frequency of follow-ups based on available data) and a brief discussion of indicators of reversible conditions in the various stages (so as to recognise and treat them in a timely manner).
- A conclusion paragraph with key take-home messages for readers is missing.
- The cited bibliography is missing, I kindly ask the Authors to add it.
Author Response
LETTER 2.
We thank the Reviewers for the thorough comments that allowed us to improve our manuscript. Answers and changes within this letter have been reported in bold, while those in the paper have been marked in yellow.
R2: The Authors conducted a literature review on the neuropsychological characterisation of the continuum between asymptomatic neuropathological changes and full-blown dementia. It is a very interesting work that, with some modifications, will provide insight and useful suggestions to clinicians and researchers interested in the topic. I have some suggestions for the authors:
- Figure 1 would be more useful if it provided, for each category, elements that would highlight the differences between each in a clear and direct way for the reader (also in the form of a bulleted list);
We thank Reviewer 2 for this suggestion. We have added the following new figure (Fig. 2, lines 90-100) with a few bullet points below each neurocognitive stage, specifically:
- Preclinical: CSF; PET; MRI biomarkers
- Transitional: cognitive complaints (patients + caregivers)
- Prodromal: cognitive-functional scores
- Major NCD: insight
- CIND (consensus on this topic is lacking, but its definition encompasses the “no dementia” stages).
Figure 2: Graphical representation of the markers or indices covered by assessment procedures over the neurocognitive continuum. The figure summarizes selected measures and information that can be collected to support diagnosis and prognosis according to the most widely adopted classification models.
Abbreviations: AD = Alzheimer’s disease; CSF = cerebrospinal fluid; CIND = cognitive impairment no-dementia; MRI = Magnetic Resonance Imaging; NCD = neurocognitive disorders; PET = Positron Emission Tomography.
We specify that each point displayed in the figure is reasoned to be cumulative to the others, meaning that all the indices reported should be addressed while dealing with a major NCD, and their collection progresses from biomarkers applicable since the preclinical stage until overt dementia, where in addition to biomarkers, cognitive complaints, and cognitive-functional scores, patients’ insight should be screened as anosognosia may challenge the collection of information about both subjective complaints and inability to perform daily activities.
R2: On the Methods section, I would advise the Authors to provide some more information on how the research was conducted: which keywords were searched? How many Authors took part in the data collection and discussion?
We thank Reviewer 2 for this request for clarification. We specify that, being a narrative review, our work did not rely on a systematic string query but on the information retrieval procedures we reported in the Methods as follows:
To yield a stadial progression of neurocognitive decline, a review of the most widely used criteria for NCDs [1,9] and MCI [10–12] was conducted. The search was al-so implemented using the online databases PubMed and Scopus to investigate the neurocognitive stages characterized by signs and clinical manifestations preceding the on-set of mild and/or major NCDs. In addition to considering their epidemiological and clinical features, the diagnostic criteria and stadial models for NCDs have also been critically discussed in relation to their applicability in primary and secondary care. All materials are available on Open Science Framework (OSF) on https://osf.io/84wnp/.
Accordingly, a combination of search terms (including, but not limited to cognitive impairment*, MCI, neurocognitive disorder*, dement*, stage*, diagnos*) was run into the two databases to expand the knowledge gathered from manuals and guidelines on NCDs. Three authors contributed to data collection and discussion, supervised by their Professors as per CRediT statement on authorship.
R2: Lines 96-97: Please specify what you mean by "very mild cognitive and/or behavioural symptoms". If the next stage (i.e. Transitional stage) includes patients with subjective problems that cannot be identified with commonly used neuropsychological tests, how are cognitive and/or motor symptoms identified and classified in this stage 1?
This point raised by Reviewer 2 is extremely important, as over the neurocognitive continuum, stage 1 is mainly asymptomatic and characterized by hallmarks for the onset of decline. In our approach, we made an effort to classify the stages on a temporal-clinical basis, trying to condense different perspectives from the literature, starting with the DSM and Petersen’s criteria, which contemplate “very mild” conditions without operationalizing them.
We have thus referred to Jessen’s work (2020) on subjective cognitive decline (SCD) which tries to define subjective complaints also based on their persistence over time. This framework requires longitudinal (e.g., 5 years) data collection to infer the presence of perceived symptoms not reflected by poorer cognitive performances. Subjective or very subthreshold symptoms may vary over a few years without even meeting the new specifications for SCD.
Moreover, our review highlights the different aetiologies and manifestations of NCDs, which may arise as noncognitive symptoms. Even in these cases, information might be collected from patients and caregivers (as for SCD), but their assessment is extremely difficult at this stage, in the absence of both a clear demarcation with the onset of stage 2 and applicable tests to capture similar cognitive-behavioral hallmarks.
Since our stages constitute a flexible, preliminary approach to the neurocognitive continuum that still requires clinical validation (as reported in the introduction), we have tried to extend the clinical focus to situations that may temporally co-occur with the onset of early biological hallmarks and evolve into stage 2 (e.g., persistent SCD) or even revert to baseline functioning. Guidance for the accurate detection of these early conditions is still required, but we contemplate the onset of extremely mild symptoms even before specific criteria for SCD are met.
R2: Sentence line 97 to 99: Please add a citation.
We thank Reviewer 2 for this comment and specify that the reference for this sentence is the same we reported for the next sentence (15), i.e., Palmqvist S, Mattsson N, Hansson O, for the Alzheimer’s Disease Neuroimaging Initiative. Cerebrospinal fluid analysis detects cerebral amyloid-β accumulation earlier than positron emission tomography. Brain. 2016;139(4):1226-1236. doi:10.1093/brain/aww015.
R2: Sentence line 115 to 117: The Authors cite PET-tau as an additional criterion for the diagnosis of AD. According to which diagnostic criteria? Its use is often almost exclusively reserved for research protocols.
We acknowledge the scarce applicability and availability of tau-PET imaging due to high costs and the limited presence of devices and specialistic competences within the clinical setting. Tau pathology is indeed an additional criterion for AD diagnosis which is primarily used in research protocols based on the A/T/N model (Jack et al., 2018, cited in the article) and not routinely adopted in standard clinical practice. However, an increasing number of centers are collecting such data, which proved being very useful in discriminating the etiology underlying neurocognitive disorders as amyloidosis (i.e., “A”) and neurodegeneration (i.e., “N”), taken alone, do not constitute the “AD signature”. According to the aforementioned research team from Mayo Clinic, the inclusion of the Tau (i.e., “T”) criterion contributes to an unbiased and comprehensive classification of the disease, also usable for clinical purposes to better understand the expression of clinical symptoms in a more precise way compared to amyloid PET data. Although “A”, “T”, and “N” biomarkers are rarely available as a set, these biomarkers may be very informative in healthcare. Indeed, professionals are becoming increasingly aware about their importance, compared to that of other signs that may be associated with AD diagnosis but still are not specific and/or do not represent established biomarkers. Therefore, we agreed about including this reference in the paper because: 1) it may represent an open, ongoing challenge pertaining to the use of evidence-based criteria in specialized care; 2) the role of AD biomarkers is also being studied in clinical research about dementing disorders other than Alzheimer pathology.
R2: Lines 118-119: Please add some details on what is meant by AD atypical variants and what their hallmarks are from the earliest stages, citing relevant literature on the subject.
We thank Reviewer 2 for this valuable input. We made further research on AD atypical forms we cited while addressing the signs for AD signature and added a reference to gather more specific knowledge. In particular, we added the following sentence (lines 151-153):
However, MRI findings show that neurodegeneration in atypical forms of AD may primarily cluster in brain areas other than medial temporal regions (e.g., posterior cortical or frontoparietal atrophy; Graff-Radford et al., 2021).
No specific addition has been made to address tau PET as “longitudinal tau accumulation occurs in frontal regions in [both] atypical variants and typical Alzheimer’s disease”, and CSF concentrations as “few studies have directly compared the profiles of typical and atypical Alzheimer’s Disease. CSF differences might include increased tau in atypical phenotypes, with mixed evidence of whether p-tau differs between variants”, implying almost the same that we provided for professionals in the paper discussions. Regarding neuropsychological assessment, there are no specific implications since assessment of visual, linguistic, executive, behavioral/praxic-motor skills is an ordinary part of the comprehensive assessment protocols we mentioned.
Reference:
Graff-Radford, J., Yong, K. X., Apostolova, L. G., Bouwman, F. H., Carrillo, M., Dickerson, B. C., ... & Murray, M. E. (2021). New insights into atypical Alzheimer's disease in the era of biomarkers. The Lancet Neurology, 20(3), 222-234.
R2: I advise the Authors, especially in the sections on the early stages, when the level of cognitive functioning often does not justify the use of invasive diagnostic means, to provide information on plasma biomarkers of the various neurodegenerative diseases (e.g. neurofilament light chains, plasma amyloid) and neurophysiological markers (e.g. diagnostic and/or prognostic approaches with NIBS techniques). For example, in the transition stage, which biomarkers are commonly found on plasma and CSF in different neuropathological conditions? How can they help predict evolution (or not) to later stages?
We sincerely appreciate the Reviewer’s 2 feedback on this point. We totally agree that the use of invasive diagnostic tools is not justified in every stage of the neurocognitive continuum. As a matter of fact, there are several promising techniques (e.g., SIMOA, RT-Quic, EEG) and hallmarks of DNCs that can be adopted, although fewer constitute widely accepted and used biomarkers to date with expandable clinical translability. To clarify the potential progression of the decline, we have included an image (Fig. 2) that illustrates some aspects to be checked during pathological development.
We also have remarked the importance of a proactive, integrated approach to screening, monitoring, and treatment, and included references to non-invasive assessment tools, acknowledging their predominant use in research contexts. The value of neuropsychological assessment as a non-invasive method to monitor the neurological functioning of patients was referenced too. However, while focusing on the practical matters for professionals, it should be noted that the diagnosis, albeit based on cognitive-functional scores that may also help to discriminate neurocognitive phenotypes (e.g., “AD-like” impairments) and selectively target intervention, also requires appropriate medical-clinical examinations to understand the probable/possible underlying causes.
In view of these considerations and Reviewer’s 2 valuable suggestions, an additional paragraph (quoted on the final page of this letter) was included in the discussion part with summary references to all these strategies (lines 399-425).
R2: Prodromal or mild stage (i.e., stage 3): at this stage, what are the neuropsychological information (including neuropsychological tests of specific utility) and biomarkers (CSF, plasma and PET) that can help predict evolution into the various forms of dementia?
We thank reviewer 2 for the interesting comment. While addressing multifaceted clinical entities such as the MCI, there is no “one-size-fits-all” but individualized approaches to be adopted with patients while administering a comprehensive neuropsychological protocol. The DSM-5-TR and further literature cited (e.g., Petersen et al.’s guidelines; Costa et al.’s European consensus document) suggest using tests covering all the core cognitive-functional domains, the scores of which are typically corrected by age, education and, where possible, gender. Likewise, we acknowledge the importance of biomarkers throughout the neurocognitive continuum, especially at stages preceding the onset of a major NCD. The new figure added (i.e., Fig. 2) contributes to highlighting the “cumulative” approach professionals should follow while collecting markers and indices since the preclinical phase. We also reported in a new paragraph (lines 399-425) that prognosis, as well as diagnosis, can benefit from integrated methodologies that account for both neuropsychological and medical-clinical outcomes, citing relevant literature in which the role of biomarkers and different monitoring/treatment strategies are debated from a proactive perspective. Here are some examples of references added, addressing different approaches to MCI prognosis:
Chowdhary, N., Barbui, C., Anstey, K. J., Kivipelto, M., Barbera, M., Peters, R., ... & Dua, T. (2022). Reducing the risk of cognitive decline and dementia: WHO recommendations. Frontiers in neurology, 12, 765584.
De Caro, M., Taurisano, P., Calia, C., & Abbatantuono, C. (2022). Profili e Modelli Neuropsicologici delle patologie neurodegenerative. Franco Angeli.
Huang, X., Zhao, X., Li, B., Cai, Y., Zhang, S., Yu, F., & Wan, Q. (2021). Biomarkers for evaluating the effects of exercise interventions in patients with MCI or dementia: A systematic review and meta-analysis. Experimental Gerontology, 151, 111424.
Livingston, G., Huntley, J., Sommerlad, A., Ames, D., Ballard, C., Banerjee, S., ... & Mukadam, N. (2020). Dementia prevention, intervention, and care: 2020 report of the Lancet Commission. The Lancet, 396(10248), 413-446.
Stern, Y. (2021). How can cognitive reserve promote cognitive and neurobehavioral health?. Archives of Clinical Neuropsychology, 36(7), 1291-1295.
Song, S., Stern, Y., & Gu, Y. (2022). Modifiable lifestyle factors and cognitive reserve: A systematic review of current evidence. Ageing Research Reviews, 74, 101551.
R2: Line 224: please add the full name of the cited source as [1].
We thank reviewer 2, we named the DSM-5 explicitly.
R2: Lines 259-261: the meaning of this sentence is not clear to me, please rephrase.
We thank Reviewer 2. We rephrased this sentence as follows (lines 292-294):
The mean survival time for patients diagnosed with AD is higher, up to 7-12 years depending on the disease onset, compared to individuals with mild-to-severe dementia stages assessed through the CDR® (approximately 3-3.5 years).
R2: Lines 264-265: mortality and incidence are two different epidemiological measures. Mortality may decrease due to better medical management of the patient, but the incidence (according to
many epidemiological estimates and evaluations) of neurodegenerative dementias (especially
- AD) is likely to increase significantly over the next decades. I suggest that the Authors justify this
sentence by citing the relevant bibliography or rephrase this sentence.
We thank Reviewer 2 for the valuable feedback. We appreciate this input and agree about the fact that mortality and incidence are distinct epidemiological measures. Moreover, International organizations and research centers claim dementia cases reflect aging trend of the world’s population. We have thus revised the sentence focusing on mortality (lines 296-299) to enhance clarity and prevent potential misunderstandings regarding the topic:
The reduced NCD mortality emerging from the last decade of research might be due to: (1) a greater control of risk factors, notably in high-income countries; (2) the overall effectiveness of secondary and tertiary prevention measures or therapies available for secondary neurodegenerative diseases.
We also removed references to incidence from the following sentence consistently.
R2: In the Discussion, I suggest adding specific advice on the optimal course of action in each stage examined (e.g. preventive measures, frequency of follow-ups based on available data) and a brief discussion of indicators of reversible conditions in the various stages (so as to recognise of this review unless otherwise stated and treat them in a timely manner).
We thank Reviewer 2 for this suggestion. On the one hand, it is crucial to capture early signs of cognitive decline by monitoring patients through highly sensitive tests; on the other hand, retest procedures mostly rely on the specific instructions for their administration.
For instance, to monitor patients with Multiple Sclerosis over time, the Rao’s BRB has been designed and used for repeated administration, whereas other tools such as the RBANS may be particularly useful for a quick and time-close assessment of MCI individuals. These protocols provide patients with part A (test) and B (retest) to prevent overexposing or learning effect affecting the cognitive performance. For the same reason, some of the most widely used verbal memory tests encompass different wordlists. However, in view of the heterogeneity and flexibility of neuropsychological procedures adopted within the clinical settings (e.g., Costa et al., 2017; De Caro et al., 2022; Donders, 2020), it should be noted that there are not strict rules for follow-up and monitoring, especially if professionals are looking for early manifestations of neurocognitive disorders, even if some sources report a follow-up on yearly basis, that may also apply to relevant clinical examinations (e.g., Nation et al., 2019).
References:
Costa, A., Bak, T., Caffarra, P., Caltagirone, C., Ceccaldi, M., Collette, F., ... & Cappa, S. F. (2017). The need for harmonisation and innovation of neuropsychological assessment in neurodegenerative dementias in Europe: consensus document of the Joint Program for Neurodegenerative Diseases Working Group. Alzheimer's research & therapy, 9, 1-15.
De Caro, M., Taurisano, P., Calia, C., & Abbatantuono, C. (2022). Modelli e profili neuropsicologici delle patologie organiche e neurodegenerative. Franco Angeli.
Donders, J. (2020). The incremental value of neuropsychological assessment: A critical review. The Clinical Neuropsychologist, 34(1), 56-87.
Nation, D. A., Ho, J. K., Dutt, S., Han, S. D., Lai, M. H., & Alzheimer’s Disease Neuroimaging Initiative. (2019). Neuropsychological decline improves prediction of dementia beyond Alzheimer’s disease biomarker and mild cognitive impairment diagnoses. Journal of Alzheimer's Disease, 69(4), 1171-1182.
We tried to briefly account for these considerations while discussing prevention and monitoring strategies, given that: 1) the new figure added in the paper (Fig. 2) already contains specific bullet points for the key factors to be considered for each stage; 2) directions from the literature retrieved were not provided for each stage, so we tried to be as comprehensive and cautious as possible while discussing following recommendations for practice (lines 399-425):
Overall, the present stadial framework for NCDs may complement existing categorical diagnoses to offer practical benefits in the context of both primary and specialized care. This approach promotes early intervention through the identification of established hallmarks and symptoms that may convert to dementia, remain stable, or revert to a status of cognitive-functional wellbeing over time. Although no consensus on CIND makes it even harder to predict the prognosis of different “no dementia” syndromes, some proactive strategies can be adopted to counter mild-to-major conversion. Based on epidemiological data on the potential remissions of mild symptoms up to the prodromal stage, it becomes crucial to target the modifiable factors of dementia (e.g., cardiovascular health and lifestyle choices) aimed at fostering patients’ functioning and quality of life. While medications and cognitive-enhancing drugs can be effective in managing specific symptoms (Frederiksen et al., 2020; Pizzi et al., 2020), favorable outcomes in the early stages of the neurocognitive continuum may be partially ascribable to brain and cognitive reserve, which act as protective factors against both age- and disease-related decline (Livingston et al., 2020; Song et al., 2022; Stern et al., 2020, 2021). To this end, the involvement of caregivers since the onset of very mild symptoms is key. Caregivers can indeed engage the patients in stimulating experiences as well as monitor the persistence and impact of symptoms that may become evident at neuropsychological follow-ups. Their role in informal care, which is being increasingly recognized by institutional care teams, is equally important when patients’ functioning and insight begin to decline (Reckrey et al., 2021).
By categorizing cognitive decline into stages, the healthcare system can also allocate resources efficiently, directing support to individuals at higher risk since the preclinical phase, and evaluating cases where undergoing invasive exams can be avoided and integrated treatment strategies (Chowdhari et al., 2020; De Caro et al., 2022, Huang et al., 2021; Pini et al., 2020; Pizzi et al., 2020) can be advised. Moreover, this framework contributes to overtreatment and stigma reduction by acknowledging cognitive impairment as a continuum, encouraging individuals to seek medical help earlier and participate actively in their care decisions, and mitigating concerns that usually occur when undergoing screening (Gruters et al., 2021).
References:
Chowdhary, N., Barbui, C., Anstey, K. J., Kivipelto, M., Barbera, M., Peters, R., ... & Dua, T. (2022). Reducing the risk of cognitive decline and dementia: WHO recommendations. Frontiers in neurology, 12, 765584.
De Caro, M., Taurisano, P., Calia, C., & Abbatantuono, C. (2022). Profili e Modelli Neuropsicologici delle patologie neurodegenerative. Franco Angeli.
Frederiksen, K. S., Cooper, C., Frisoni, G. B., Frölich, L., Georges, J., Kramberger, M. G., ... & Waldemar, G. (2020). A European Academy of Neurology guideline on medical management issues in dementia. European journal of neurology, 27(10), 1805-1820.
Gruters, A. A., Christie, H. L., Ramakers, I. H., Verhey, F. R., Kessels, R. P., & de Vugt, M. E. (2021). Neuropsychological assessment and diagnostic disclosure at a memory clinic: A qualitative study of the experiences of patients and their family members. The Clinical Neuropsychologist, 35(8), 1398-1414.
Huang, X., Zhao, X., Li, B., Cai, Y., Zhang, S., Yu, F., & Wan, Q. (2021). Biomarkers for evaluating the effects of exercise interventions in patients with MCI or dementia: A systematic review and meta-analysis. Experimental Gerontology, 151, 111424.
Livingston, G., Huntley, J., Sommerlad, A., Ames, D., Ballard, C., Banerjee, S., ... & Mukadam, N. (2020). Dementia prevention, intervention, and care: 2020 report of the Lancet Commission. The Lancet, 396(10248), 413-446.
Pini, L., Manenti, R., Cotelli, M., Pizzini, F. B., Frisoni, G. B., & Pievani, M. (2019). Non-invasive brain stimulation in dementia: a complex network story. Neurodegenerative Diseases, 18(5-6), 281-301.
Pizzi, S. D., Granzotto, A., Bomba, M., Frazzini, V., Onofrj, M., & Sensi, S. L. (2020). Acting before; a combined strategy to counteract the onset and progression of dementia. Current Alzheimer Research, 17(9), 790-804.
Reckrey, J. M., Boerner, K., Franzosa, E., Bollens-Lund, E., & Ornstein, K. A. (2021). Paid caregivers in the community-based dementia care team: Do family caregivers benefit?. Clinical Therapeutics, 43(6), 930-941.
Stern, Y., Barnes, C. A., Grady, C., Jones, R. N., & Raz, N. (2019). Brain reserve, cognitive reserve, compensation, and maintenance: operationalization, validity, and mechanisms of cognitive resilience. Neurobiology of aging, 83, 124-129.
Stern, Y. (2021). How can cognitive reserve promote cognitive and neurobehavioral health?. Archives of Clinical Neuropsychology, 36(7), 1291-1295.
Song, S., Stern, Y., & Gu, Y. (2022). Modifiable lifestyle factors and cognitive reserve: A systematic review of current evidence. Ageing Research Reviews, 74, 101551.
R2: A conclusion paragraph with key take-home messages for readers is missing.
We thank Reviewer 2 for this advice. We added the following concluding paragraph (lines 428-438):
The recognition and monitoring of individuals at stages from preclinical to overt dementia are essential to optimize clinical efforts against neurocognitive decline. These strategies may allow healthcare professionals to assess the progression of cognitive decline stepwise, orient care plans to evolving patient needs, and provide timely interventions to enhance their quality of life. In view of the validation of further assessment techniques and dementia biomarkers, the adoption of a perspective that accounts for clinical-temporal progression can enrich the diagnostic systems and supportive measures currently in use, ultimately advocating wellbeing and independence for individuals experiencing neurocognitive symptoms.
R2: The cited bibliography is missing, I kindly ask the Authors to add it
We thank Reviewer 2 for notifying this issue. We actually noticed that the references were not listed by the journal in the PDF version, but they are visible in the Word version. We checked the old and new references which were reported in AMA format using Zotero reference manager. We could visualize them correctly in the manuscript Word file. However, we remain fully available to make further changes to enhance our reference list if required.

Round 2
Reviewer 2 Report
The Authors appropriately responded to comments and suggestions. The paper is now, in my opinion, suitable for publication